# Adverse Pregnancy Outcomes after Multi-Professional Follow-Up of Women with Systemic Lupus Erythematosus: An Observational Study from a Single Centre in Sweden

**DOI:** 10.3390/jcm9082598

**Published:** 2020-08-11

**Authors:** Muna Saleh, Christopher Sjöwall, Helena Strevens, Andreas Jönsen, Anders A. Bengtsson, Michele Compagno

**Affiliations:** 1Department of Biomedical and Clinical Sciences, Division of Inflammation and Infection, Linköping University, SE-581 85 Linköping, Sweden; christopher.sjowall@liu.se; 2Department of Clinical Sciences Lund, Department of Obstetrics and Gynaecology, Lund University, SE-222 42 Lund, Sweden; helena.strevens@med.lu.se; 3Department of Clinical Sciences Lund, Rheumatology, Lund University, SE-222 42 Lund, Sweden; andreas.jonsen@med.lu.se (A.J.); anders.bengtsson@med.lu.se (A.A.B.); michele.compagno@med.lu.se (M.C.)

**Keywords:** SLE, pregnancy, conception, adverse pregnancy outcomes, maternal and foetal complications, Lupus nephritis, antiphospholipid syndrome, risk factors

## Abstract

While the management of pregnant patients with systemic lupus erythematosus (SLE) has improved over the last decades, the risk of maternal, foetal, and neonatal complications is still substantial. We evaluated the occurrence of adverse pregnancy outcomes (APO) occurring in 2002–2018 among patients with SLE from the catchment area of the Department of Rheumatology in Lund, Sweden. Longitudinal clinical and laboratory data were collected and analysed. Results were stratified according to the sequence of conception. We investigated a total of 59 pregnancies in 28 patients. Prior lupus nephritis was the clinical feature that, in a multivariable regression analysis, displayed the strongest association with APO overall (OR 6.0, *p* = 0.02). SLE combined with antiphospholipid syndrome (APS) was associated with the risk of miscarriage (OR 3.3, *p* = 0.04). The positivity of multiple antiphospholipid antibodies (aPL) was associated with APO overall (OR 3.3, *p* = 0.05). IgG anti-cardiolipin during pregnancy resulted in a higher risk of preterm delivery (OR 6.8, *p* = 0.03). Hypocomplementaemia was associated with several APO, but only in the first pregnancies. We conclude that, despite the close follow-up provided, a majority of pregnancies resulted in ≥1 APO, but a few of them were severe. Our study confirms the importance of previous lupus nephritis as a main risk factor for APO in patients with SLE.

## 1. Introduction

Systemic lupus erythematosus (SLE) is a systemic autoimmune disease that often affects fertile women. Adverse pregnancy outcomes (APO), commonly grouped into maternal and foetal/neonatal complications, occur in the obstetric general population. Pregnant SLE patients are at increased risk of both maternal and foetal/neonatal APO [1,2]. In the past, patients with SLE have been advised and warned from getting pregnant and giving birth, due to the high risk of severe complications for both the mother and the offspring. In recent times, modern treatment strategies and established preventive measures have led to less risk of APO, but there is more to be done to keep the risk as low as in the obstetric population.

Maternal APO observed in SLE include increased disease activity, pre-eclampsia, eclampsia and Haemolysis, Elevated Liver enzymes and Low Platelet count (HELLP) syndrome, as well as obstetrical complications, such as preterm labour, unplanned Caesarean delivery, and conditions related to pre-eclampsia [3,4,5]. Foetal and neonatal complications frequently associated with SLE are miscarriage, stillbirth, intrauterine growth restriction (IUGR), neonatal lupus erythematosus (NLE), congenital heart block, and prematurity [6,7,8]. Moreover, women with SLE are at increased risk of foetal loss around the 10th gestational week, particularly in the presence of active SLE, active lupus nephritis (LN), or concomitant antiphospholipid syndrome (APS) [9]. However, a decline in the risk of foetal loss in SLE has been reported over the last decades [9,10,11,12]. Up to 30% of pregnancies in SLE patients are complicated by IUGR and small-for-gestational-age (SGA) newborns, compared to approximately 10% of pregnancies in the general obstetric population [13,14]. Lower birth weight at any gestational age is also more prevalent among the offspring of SLE patients [15]. Pre-eclampsia is one of the most frequent maternal APO in SLE, occurring in 16% to 30%, compared with 5% of pregnancies in the general obstetric population [16,17]. Poorly controlled SLE, a history of LN, low levels of complement proteins, and thrombocytopenia have been identified as risk factors for pre-eclampsia in SLE [16,18,19]. The role played by the antiphospholipid antibodies (aPL) in the occurrence of pre-eclampsia is still controversial [16,19]. Active SLE, previous LN, presence of APS or aPL, and thrombocytopenia have been suggested as predictors, as well as putative risk factors of other APO [16,20,21,22], but studies are needed to evaluate this further.

The study aimed to contribute to the field of knowledge with a report of real-world longitudinal data from one single tertiary referral rheumatology centre in southern Sweden, where a multi-professional follow-up of pregnant SLE patients has been carried out since 2002. In particular, we aimed to assess the occurrence of APO in our cohort and how some established and putative SLE-related risk factors may predict the pregnancy outcomes.

## 2. Methods

### 2.1. Patients and Clinical Follow-Up

In patients affected by SLE, from the catchment area of Skåne University Hospital, Sweden, we investigated the outcome of all the pregnancies occurred between 2002 and 2018. Each included patient fulfilled the 1997 American College of Rheumatology (ACR) and/or the 2012 Systemic Lupus International Collaborating Clinics (SLICC) classification criteria [23,24] and was regularly followed at the Department of Rheumatology, since the time-point of diagnosis. All patients had consented to join a prospective follow-up program, intending to improve care of SLE patients and to identify clinical and laboratory features that could be used as a marker or a predictor of complications and exacerbations of the disease. Patients were followed longitudinally with scheduled visits every 60 ± 20 days and extra visits in case of disease flares. An extensive set of clinical and laboratory variables were registered in a database tailored for the study. Serum samples were collected regularly, also before and after the date of clinical assessment.

Whenever pregnant, the patients underwent multi-professional follow-up, with recurrent visits at the local antenatal clinic, often in presence of one of the health professionals responsible for the regular follow-up at the rheumatology department. After informed consent from the pregnant patients, we gathered the relevant clinical features and laboratory data already recorded in the local SLE database, to define the clinical phenotype of each patient [25], as well as the data recorded during each and every gestation that occurred during the study period, including the medical records from obstetrics and neonatal units concerning all the maternal and perinatal outcomes. Concomitant APS, meeting the Sydney classification criteria [26], was accounted for only if APS had been recognized before conception.

Disease activity and acquired organ damage were assessed using the SLEDAI-2000 (SLEDAI-2K) [27] and the SLICC/ACR damage index (SDI) [28], respectively. A SLEDAI-2K score of ≥4 was recorded as “active SLE”. To the purpose of this study, we gathered the SLEDAI-2K scores determined at 6-months before conception, at each trimester, and at 6-months post-pregnancy, as well as the SDI scores determined at 1 year before conception and at 1 year after termination of each gestation.

To account for systematic bias, we analysed all the first pregnancies occurring after the diagnosis of SLE apart from the remaining pregnancies. Women who had been pregnant up to 12 months before the time-point of SLE diagnosis were not suitable for the “first pregnancy group” but for the “subsequent pregnancy” group. Identical statistical analyses were employed separately in these two subgroups, as well as in the pooled data of all pregnancies.

### 2.2. APO

We collected data concerning the following maternal complications, documented during the investigated pregnancies: gestational hypertension, pre-eclampsia (or eclampsia or HELLP syndrome), preterm delivery (<37th gestational week), and gestational diabetes.

We assessed the occurrence of the following foetal/neonatal complications: foetal loss, defined as early miscarriage (occurring before 10th week of gestation), or late miscarriage (occurring between the 10th and 24th week of gestation), or intrauterine foetal death (IUFD) (occurring at >24th gestational weeks); stillbirth; IUGR (a foetus not reaching its target weight based on sonographic estimated foetal weight); SGA newborns (weight and length at birth below 10th percentile for gestational age); low birth weight (LBW-below 2500 gram, regardless of gestational age); congenital heart block and NLE.

### 2.3. Risk Factors

We investigated the role of SLE-related clinical and immunological features as putative risk factors for the development of APO. The following clinical features were assessed and gathered as binary variables (yes/no): concomitant APS (recognized before the time of conception) and history of LN (documented by renal biopsy). In addition, acknowledged risk factors for APO in the general obstetric population were studied, such as smoking habits (tobacco smoking; ever or current), obesity (body mass index >30) and age ≥35 years at the time of conception [29].

The following immunological findings were gathered as binary variables (yes/no) and grouped as “ever present”, “present 1-year before conception” and “present during pregnancy”: presence of anti-dsDNA, anti-Ro/SSA, anti-La/SSB, anti-cardiolipin (aCL, IgG isotype), anti-β2-glycoprotein-I (anti-β2GPI, IgG isotype), lupus anticoagulant (LA) test and hypocomplementaemia (decreased levels of complement proteins: C3 and/or C4 and/or C1q).

The assessments were made at the local clinical immunology laboratory, according to the validated assays and methods in current use at the time of evaluation.

### 2.4. Statistics

Statistical analyses were performed with SPSS Statistics, version 25.0 (IBM, Armonk, NY, USA) and GraphPad Prism, version 6.07 (GraphPad Software, La Jolla, CA, USA). Median values were calculated for continuous variables. Other variables were presented as binary categorical variables, apart from the SLEDAI-2K and SDI scores, collected as continuous variables and presented (mean ± SD) in a Box-whisker plots graph. To handle with repeated measurements during each gestation and with data collected from recurrent gestations in the same patient, we used Generalized Estimating Equation (GEE) to assess the association between potential risk factors and outcomes (APO). A *p*-value of <0.05 was considered significant. Potential risk factors for developing any APO in each studied pregnancy were examined also by logistic regression analysis (univariate and multivariable). All the significant associations in the univariate model were included in the multivariable analysis. No corrections for multiple comparisons were made, but by reporting the exact *p*-values, we enable this by any preferred method [30].

### 2.5. Ethical Considerations

Oral and written informed consent was obtained from all participants. The study protocol was approved by the regional ethics board in Lund (Dnr: LU 378–02).

## 3. Results

Twenty-eight patients affected by SLE experienced 1–5 gestations each during the study period. We totally investigated 59 pregnancies, whereof 26 were categorised as “1st pregnancies”. The remaining 2 patients had had ≥1 gestation before the onset of SLE; therefore, only their subsequent pregnancies were recorded. Twenty-one patients accounted for the 33 “subsequent pregnancies”. Overall, 61 embryos were conceived, being 2 out of the 59 gestations multiple (3.4%) with twins.

All recorded demographic data, clinical manifestations, immunological features, and pharmacological therapy concerning the included patients are summarized in Table 1 and Table 2.

### 3.1. APO

At least 1 APO complicated 33 pregnancies (56%) in 18 patients (64%). A total of 44 gestations (75%) ended with delivery, whereof 8 (18%) were preterm. No post-term deliveries were recorded. As shown in Table 3, pre-eclampsia (25%), preterm deliveries (18%), and Caesarean section (30%) were the most common maternal APO.

Regarding foetal/neonatal adverse outcomes, we observed 13 cases (22%) of early foetal loss, but no cases of late miscarriage, IUFD, or stillbirth. Restricted foetal growth (IUGR and/or, SGA, and/or LBW) was recorded in 13 pregnancies (30%). One case (1.7%) of NLE and no congenital heart block was observed.

Among the 26 “1st pregnancies”, 22 (85%) ended with delivery and 4 with early miscarriage. A total of 11 of these 22 deliveries (50%) were complicated with ≥1 APO. The 33 “subsequent pregnancies” resulted in 22 (66%) deliveries, 9 (27%) early miscarriage and 2 (6%) induced abortions. Seven deliveries (32%) were complicated with ≥1 APO. Further details concerning types of APO observed are summarized in Table 3. The incidence of APO over time decreased around 10%, despite about 2/3 of the investigated pregnancies were recorded during the period 2010–2018, as illustrated in Table 4.

### 3.2. Risk Factors

The analysis of all the 59 gestations showed that APS diagnosed before the time of conception in SLE patients may enhance the risk (OR 3.3, *p* = 0.04) of early miscarriage. Furthermore, the presence of aPL was associated with increased risk (OR 3.3, *p* = 0.05) of any APO. In particular, the presence of aCL during pregnancy resulted in a higher risk (OR 6.8, *p* = 0.03) of preterm delivery, regardless of APS. A history of LN was associated with increased risk of any APO (OR 5.9, *p* = 0.005), particularly any kind of restricted foetal growth (OR 16.6, *p* = 0.01).

At first pregnancy after the onset of SLE, the detection of aCL during the gestation was associated with an increased risk of preterm delivery (OR 32.0, *p* = 0.03). Restricted foetal growth was associated with presence of different immunological features and clinical manifestations, such as low levels of C3 (OR 13.0, *p* = 0.04) and C4 (OR 20.0, *p* = 0.02) up to 1-year before conception, as well as presence of aPL during pregnancy (OR 17.3, *p* = 0.03), concomitant APS diagnosis (OR 13.0, *p* = 0.04) and previous LN (OR 12.6, *p* = 0.04). In addition, increased risk of pre-eclampsia/HELLP was associated with low C4 levels detected up to 1-year before conception (OR 12.5, *p* = 0.04). 

In the subgroup of 33 subsequent pregnancies, a positive aCL test ≤1-year before the time of conception (OR 3.8, *p* = 0.01), concomitant APS diagnosis (OR 4.9, *p* = 0.02) and a history of LN (OR 5.7, *p* = 0.04) indicated higher risk of miscarriage. Ever documented presence of aPL (OR 11.3, *p* = 0.04) and previous LN (OR 12.0, *p* = 0.01) were also associated with increased risk for any APO. In contrast, ever documented low C3 levels (OR 0.05, *p* = 0.03) showed an inverse association with pre-eclampsia/HELLP. The associations between risk factors and APO are summarized in Table 5 and are further detailed in Appendix A.

Next, we performed a multivariable logistic regression analysis, emphasizing that previous LN was the only significant risk factor in our study population for the occurrence of any APO (OR 6.0, *p* = 0.02), particularly in the subsequent pregnancy group (OR 25.7, *p* = 0.02). In the subgroup of 26 first pregnancies, none of the investigated risk factors showed any significant associations with maternal or foetal APO in the univariate model.

### 3.3. Damage Accrual and Disease Activity

SDI and SLEDAI-2K were determined at each visit, to assess accrual of organ damage and disease activity, respectively. No major changes in SDI score were found before and after each pregnancy, regardless of the number of pregnancies and the occurrence of APO (Figure 1). Moreover, a modest increase in disease activity during the last two trimesters and the near period post-delivery was observed. Active SLE (SLEDAI-2K score of ≥4) was found in around 30% during the period before conception and during the 1st trimester of each gestation. The rate of pregnant patients with active SLE increased up to 49% and 43% during the 2nd and the 3rd trimester of gestation, respectively. We documented, at last, active SLE in 51% of the investigated pregnancies during the six months following the termination of the gestations (Figure 1).

## 4. Discussion

This is the first Swedish study that describes the occurrence of APO in a well-defined population of patients affected by SLE, undergoing regular multi-professional check-ups throughout pregnancy, at one single university centre. In previous studies, this kind of regular follow-up has been suggested to facilitate a positive outcome of pregnancy in patients with SLE, particularly those with a history of LN and concomitant APS or with the presence of aPL [9,15,20]. Despite progress in the management of pregnancies in SLE patients during the last decades [2,13], our data indicate that SLE is still associated with high risks of maternal and foetal complications. While pre-eclampsia occurs in 2–8% of pregnancies in the general obstetric population and is seldom complicated by eclampsia and the HELLP syndrome, pre-eclampsia occurred in >20% of the pregnancies investigated in the present study, which is comparable to the results reported in previous investigations [3,16,31].

Early foetal loss and preterm deliveries were also more common than usually observed in the general population [3,6,9,32]. Late foetal loss, stillbirth, and congenital heart defects are generally less prevalent [33]. The absence of these APO in our experience could simply be due to the limited number of pregnancies investigated, though the regular multi-professional follow-up may have contributed to fewer major foetal/neonatal APO in the later phase of pregnancies. IUGR occurs in about 10–30% among pregnant women with SLE, as compared to approximately 10% in the general obstetric population [4,13,15]. Around 30% of gestations in our study resulted in restricted foetal growth, such as IUGR and SGA or LBW. Among the risk factors assessed in this study, a history of LN, diagnosis of APS, and the presence of one or more aPL were associated with major APO.

Nevertheless, the results of the present investigation may suggest a trend towards a lower incidence of APO in SLE patients during the last years (54%), as compared with the incidence recorded between 2002 and 2009 (60%). We speculate that the accurate use of effective and safe drugs, as well as the implementation of a multi-professional follow-up program, has contributed to achieving a lower rate of APO among patients with SLE, despite the growing number of recorded gestations in our centre.

Renal involvement in SLE has previously been reported as a risk factor for developing several APO, mainly foetal loss, preterm delivery, restricted foetal growth, and pre-eclampsia [13,14,34,35]. Previous LN among our patients showed a significant association with the development of any APO, in particular with restricted foetal growth, as confirmed by the assessment in a multivariable regression model (Table 6). However, none had active LN at the time of conception or up to 6 months before conception. One patient developed LN during the 2nd trimester of pregnancy and needed to terminate the gestation at the 20th week.

Concomitant APS and the presence of aPL have already been reported as potential risk factors for the development of APO [20,21,36,37], and this was confirmed herein. We could confirm that abnormal levels of aCL during pregnancy indeed were associated with the occurrence of preterm delivery, as previously described [38,39]. Worth noting is that, in our population, APS appeared to be a significant risk factor for APO, despite that the diagnosis was already known since before the time of conception and the treatment was ongoing.

The results of our investigation show that hypocomplementaemia, or activation of the complement system, during the first pregnancy after the presumptive onset of SLE, may have a pathological role in the occurrence of some APO, such as pre-eclampsia and restricted foetal growth (Appendix A). This is consistent with previous investigations that indicate the activation of the classical pathway of the complement system, and hypocomplementaemia overall, as variables associated with a significant risk of APO. In particular, a higher occurrence of foetal loss, IUGR, and pre-eclampsia have been observed, especially when concomitant APS or aPL [18,40,41,42,43,44,45] are present. Moreover, the activation of the complement system has been demonstrated to be of importance regarding the risk of pre-eclampsia and IUGR, even in women without autoimmune diseases [46,47].

On the other hand, any decreased level of C3 ever documented in our patients showed an inverse association with the occurrence of pre-eclampsia/HELLP syndrome during the subsequent pregnancies. These results were unexpected and should be interpreted with caution. However, it may reflect the tailored interventions usually undertaken to minimize the risk of APO in patients with a history of hypocomplementaemia [48]. In line with this, the use of antimalarials, low-molecular-weight heparin, and acetylsalicylic acid increased during subsequent pregnancies, in comparison with first pregnancies (Table 2), which might have contributed to reducing the risk of APO.

Many reports have emphasized the importance of low disease activity in women with SLE at the time of conception and throughout the entire pregnancy period, to minimize the risk of APO [49,50,51]. Kwok et al., concluded that a SLEDAI score of ≥4 during the 6 months before conception predicts adverse maternal outcomes, while a disease flare during pregnancy rather predicts adverse foetal outcomes [51]. Similar conclusions have been drawn in other studies [49,50,51,52]. In our study population, SLEDAI-2K scores were generally low 6 months prior to conception and during the 1st trimester. The mean values of SLEDAI-2K were slightly higher during the latter part of the investigated pregnancies, as well as during 6 months post-partum, as shown in Figure 1, consistent with other investigations [53,54]. No significant increase of organ damage was recorded among our cases, which also was observed in a recently published study, suggesting that pregnancies before and after the diagnosis of SLE may not be significant predictors of irreversible damage [55]. Effects of other general risk factors for APO (i.e., smoking, ≥35 years of age at the time of conception, or obesity) did not fall out significant. However, the general risk factors among the included women were uncommon. Subsequently, firm conclusions regarding this cannot be drawn based on our data.

Some limitations should be acknowledged. Firstly, we could not investigate the effect of medication on the major APO. Moreover, the small number of studied pregnancies in a limited number of patients from one single centre may have led to uncertain results, and conclusions could thus be difficult to generalize to a broader population of SLE patients. Finally, one must also consider the risk of observation and treatment bias; patients so closely followed might run the risk of being delivered at the first signs of a developing APO, such as pre-eclampsia or restricted foetal growth, thereby increasing the number of abnormal parameters observed but limiting the number of serious APO. On the other hand, a major strength is the Swedish healthcare system, which is public, tax-funded, and offers universal access. This significantly reduces the risk of selection bias and ensures a very high coverage of cases, especially at a tertiary referral centre, offering high-specialized health-care services with longstanding experience of SLE care [25]. A clear advantage of a single centre study in comparison with a multi-centre study is the homogeneity in both care and management over time in this patient group. For this reason, though small in size, this study provides reliable data concerning appropriate management of these women.

## 5. Conclusions

To conclude, the results of the present investigation highlight the risks of APO in SLE. No new predictors could be identified in this study. Nonetheless, our data indicate the importance of planning pregnancy and organizing a multi-professional follow-up during pregnancy, with regular visits, to minimize the prevalence of APO and/or the consequences of APO. The early detection of unfavourable conditions may help to step up surveillance and to provide the patients with the best treatment to prevent and avoid serious adverse outcomes. 

The sequence of conception may play an important role in the risk stratification and prevention. We confirm the importance of some clinical phenotypes (e.g., previous LN, APS) and immunological factors (e.g., aPL, hypocomplementaemia) as risk factors for the occurrence of APO. Larger multicentre studies would be needed to identify further reliable predictors of APO and to investigate the actual impact of pharmacological treatments.

## Figures and Tables

**Figure 1 jcm-09-02598-f001:**
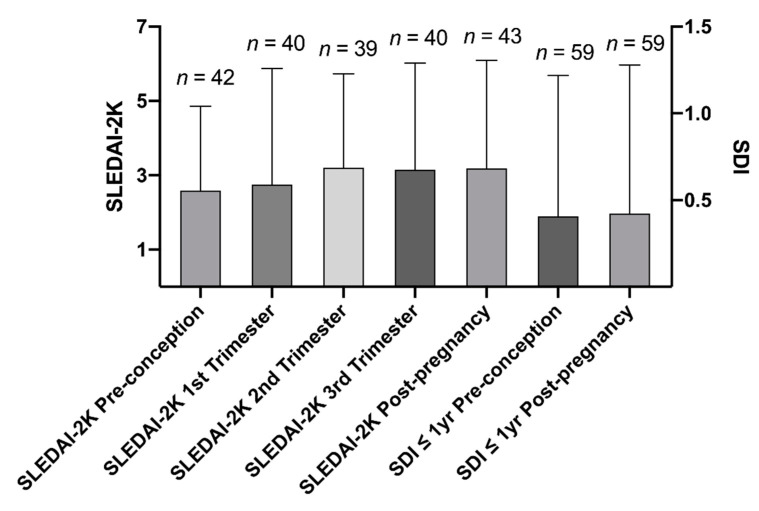
SLEDAI-2K scores illustrated during each trimester as well as 6 months pre-conception and 6 months post-pregnancy; and SLICC/ACR damage index (SDI) up to 1 year pre-conception and 1 year post-pregnancy (termination of pregnancy). Data show mean ± standard deviations.

**Table 1 jcm-09-02598-t001:** Characteristics and descriptive data of the studied pregnancies (*n* = 59) in included patients (*n* = 28).

Variable	Value	Range or Percent (%)
Caucasian race/ethnicity, *n*/total	27/28	96.4%
Age at SLE diagnosis, median (years)	20.5	10–33
Age at time of conception, median (years)	31	22–41
Disease duration at time of conception, median (years)	10	0–25
BMI at time of conception, median (kg/m^2^)	23.5	19–33
Ever smoked tobacco before conception, *n*/total	10/48	20.8%
Ever treated with antihypertensives, *n*/total	12/59	20.3%
Diabetes mellitus, *n*/total	0/59	0%
Pregnancies with prior APS, *n*/total	19/59	32.2%
**2012 SLICC criteria** [23]	**In 28 patients** (%)	**In 59 pregnancies** (%)
Acute cutaneous lupus, *n* (%)	17 (60.7)	34 (57.6)
Chronic cutaneous lupus, *n* (%)	4 (14.3)	6 (10.2)
Oral ulcers, *n* (%)	7 (25)	12 (20.3)
Non-scarring alopecia, *n* (%)	8 (28.6)	10 (16.9)
Synovitis, *n* (%)	27 (96.4)	58 (98.3)
Serositis, *n* (%)	9 (32.1)	16 (27.1)
Renal, *n* (%)	15 (53.6)	36 (61.0)
Neurologic, *n* (%)	6 (21.4)	13 (22.0)
Haemolytic anaemia, *n* (%)	3 (10.7)	7 (11.9)
Leukopenia, *n* (%)	18 (64.3)	41 (69.5)
Lymphopenia, *n* (%)	25 (89.3)	54 (91.5)
Thrombocytopenia, *n* (%)	11 (39.3)	20 (33.9)
ANA, *n* (%)	28 (100)	59 (100)
Anti-Sm antibody, *n* (%)	5 (17.9)	13 (22.0)
Anti-dsDNA antibody, *n* (%)	17 (60.7)	35 (59.3)
Antiphospholipid antibodies ***, (%) *n*	17 (60.7)	38 (64.4)
Low complement, *n* (%)	25 (89.3)	40 (67.8)
Positive direct Coombs’ test in the absence of haemolytic anaemia, *n* (%)	5 (17.9)	10 (16.9)

* With or without APS diagnosis; ANA, antinuclear antibodies; APS, antiphospholipid syndrome; BMI, Body Mass index.

**Table 2 jcm-09-02598-t002:** Pharmacotherapy and immunological features in the study population.

	Ever Documented	Up to 1 Year before Pregnancy			During Pregnancy		
Pharmacotherapy	Total (*n* = 59)	Total (*n* = 59)	FP (*n* = 26)	SP (*n* = 33)	Total (*n* = 59)	FP (*n* = 26)	SP (*n* = 33)
<15 mg prednisolone		28	14	14	30	14	16
≥15 mg prednisolone		2	2	0	9	5	4
Antimalarials	50	47	21	26	44	20	24
Azathioprine	41	31	14	17	31	14	17
Mycophenolate mofetil	19	1	1	0	0	0	0
Cyclophosphamide	15	0	0	0	0	0	0
Cyclosporine	6	0	0	0	0	0	0
Tacrolimus	3	0	0	0	0	0	0
Protein A *	3	0	0	0	0	0	0
Intravenous immunoglobulin	4	0	0	0	1	1	0
Biologics	0	0	0	0	0	0	0
Methotrexate	2	0	0	0	0	0	0
Warfarin	10	7	3	4	0	0	0
LMWH	17	1	1	0	17	6	11
Acetylsalicylic acid	18	16	7	9	34	12	22
**Immunological data**							
Anti-dsDNA, *n*/total (%)	35/59 (59.3)	12/55 (21.8)			17/51 (33.3)		
Anti-Ro/SSA, *n*/total (%)	14/59 (23.7)	12/53 (22.6)			13/51 (25.5)		
Anti-La/SSB, *n*/total (%)	13/59 (22.0)	10/53 (18.9)			12/51 (23.5)		
Anti-cardiolipin, *n*/total (%)	38/59 (64.4)	12/54 (22.2)			5/51 (9.8)		
Anti-β2-GPI, *n*/total (%)	15/59 (25.4)	8/52 (15.4)			9/51 (17.6)		
LA test, *n*/total (%)	16/54 (29.6)	4/22 (18.2)			5/41 (12.2)		
Low C3, *n*/total (%)	40/59 (67.8)	22/53 (41.5)			20/51 (39.2)		
Low C4, *n*/total (%)	38/59 (64.4)	24/53 (45.3)			21/51 (41.2)		
Low C1q, *n*/total (%)	35/53 (66.0)	12/53 (22.6)			26/51 (51.0)		

* Immune-adsorption Staphylococcal protein A; aCL, anti-cardiolipin antibody; anti-β2-GPI, anti-β2-glycoprotein I antibody; FP, first pregnancy after SLE diagnosis; IVIG, intravenous immunoglobulin; LA, lupus anticoagulant test; LMVH, low molecular-weight heparin; SP, subsequent pregnancy.

**Table 3 jcm-09-02598-t003:** Maternal and foetal adverse pregnancy outcomes (divided by pregnancy number in the case of early outcome, or by number of deliveries in the case of late outcome).

	APO in All Pregnancies (*n* = 59)	APO in FP (*n* = 26)	APO in S (*n* = 33)
**Maternal APO**			
Pre-eclampsia, *n*/total (%)	11/44 (25)	6/22 (27)	5/22 (23)
Eclampsia, *n*/total (%)	0/44 (0)	0/22 (0)	0/22 (0)
HELLP syndrome, *n*/total (%)	1/44 (2)	1/22 (5)	0/22 (0)
Preterm deliveries (<37th gestational week), *n*/total (%)	8/44 (18)	4/22 (18)	4/22 (18)
Gestational hypertension, *n*/total (%)	2/44 (5)	2/22 (10)	0/22 (0)
Gestational diabetes, *n*/total (%)	1/44 (2)	0/22 (0)	1/22 (5)
Caesarean delivery, *n*/total (%)	13/44 (30)	7/22 (32)	5/22 (23)
**Foetal/Neonatal APO**			
Miscarriage <10 weeks, *n*/total (%)	13/59 (22)	4/26 (15)	9/33 (27)
Miscarriage ≥10 weeks, *n*/total (%)	0/59 (0)	0/26 (0)	0/33 (0)
Induced abortion	2/59 (3)	0/26 (0)	2/33 (6)
IUFD >24 weeks, *n*/total (%)	0/44 (0)	0/22 (0)	0/22 (0)
Stillbirths, *n*/total (%)	0/44 (0)	0/22 (0)	0/22 (0)
Prematurity, *n*/total (%)	10/44 * (23)	5/22 (23)	5/22 (23)
Restricted foetal growth, *n*/total	13/44 (30)	8/22 (36)	5/22 (23)
IUGR, *n*/total (%)	5/44 (11)	3/22 (14)	2/22 (9)
SGA, *n*/total (%)	1/44 (2)	1/22 (0.5)	0/22 (0)
LBW, *n*/total (%)	10/44# (23)	6/22 (27)	4/22 (18)
Congenital heart block, *n*/total (%)	0/44 (0)	0/22 (0)	0/22 (0)
Neonatal lupus erythematosus, *n*/total (%)	1/44 (2)	1/22 (0.5)	0/22 (0)

* Birth before 37th week of gestation (2 multiple gestations with twins). # 2 cases already diagnosed with IUGR and 1 case also diagnosed with SGA. These gestations resulted in 15 newborns (2 multiple gestations with twins). FP, first pregnancy after SLE diagnosis; IUFD, intrauterine foetal death; IUGR, intrauterine growth restriction; LBW, low birth weight; SGA, small for gestational age; SP, subsequent pregnancies.

**Table 4 jcm-09-02598-t004:** Incidence of adverse pregnancy outcomes (APO) grouped according to the decade of occurrence. Observe that one pregnancy may have been complicated by >1 APO.

Year Interval	Gestations/Patients		Total APO *n* (%)	Pregnancies with APO (*n* = 33)		Pre-Eclampsia/HELLP (*n* = 12)		Miscarriage (<10 weeks) (*n* = 13)		Preterm Delivery (*n* = 8)		Restricted Foetal Growth * (*n* = 13)	
Sequence of conception	FP	SP		FP	SP	FP	SP	FP	SP	FP	SP	FP	SP
2002–2009	11/11	9/5	12 (60%)	6	6	1	1	2	4	2	2	3	2
2010–2018	15/15	24/16	21 (54%)	9	12	6	4	2	5	2	2	5	3
Total	26/26	33/21		15	18	7	5	4	9	4	4	8	5

* IUGR (intra-uterine growth restriction) and/or SGA (small-for-gestational-age) newborns and/or LBW (low birth weight). FP, first pregnancy after SLE diagnosis; SP, subsequent pregnancies.

**Table 5 jcm-09-02598-t005:** Associations between investigated risk factors and the APO reaching statistical significance at any time-point among all included pregnancies (59 conceptions, whereof 44 led to delivery). *p*-values, odds ratios, and 95% confidence intervals are given.

Risk Factor		Total APO (*n* = 33)	Pre-Eclampsia and HELLP (*n* = 12)	Miscarriage (<10 Weeks) (*n* = 13)	Preterm Delivery (*n* = 8)	Restricted Foetal Growth * (*n* = 13)
	Denominators	*n =* 59	*n =* 44	*n =* 59	*n =* 44	*n =* 44
**aCL**	Ever	0.40 (OR 1.7, CI 0.5–5.7)	0.51 (OR 1.8, CI 0.3–9.5)	0.08 (OR 3.9, CI 0.9–17.2)	0.31 (OR 2.7, CI 0.4–18.3)	0.83 (OR 0.8, CI 0.2–3.9)
	≤1-year before	0.74 (OR 0.8, CI 0.3–2.6)	0.33 (OR 0.3, CI 0.03–3.1)	0.47 (OR 1.4, CI 0.6–3.7)	0.12 (OR 4.8, CI 0.7–35.0)	0.61 (OR 1.5, CI 0.3-7.1)
	During pregnancy	0.46 (OR 1.6, CI 0.4–6.1)	N.E.	N.T.	**0.03 (OR = 6.8, CI 1.2–39.4)**	0.19 (OR 2.9, CI 0.6–14.1)
**≥1 pos. aCL, anti-β2-GPI or LA test**	Ever	**0.05 (OR 3.3, CI 1.0–11.11)**	0.17 (OR 3.4, CI 0.6–19.6)	0.20 (OR 2.7, CI 0.6–11.8)	0.59 (OR 1.7, CI 0.2–11.8)	0.36 (OR 2.1, CI 0.4–10.2)
	≤1-year before	0.73 (OR 0.8, CI 0.3–2.6)	0.57 (OR 0.6, CI 0.1–3.3)	0.87 (OR 1.1, CI 0.4–3.1)	0.23 (OR 3.4, CI 0.5–24.5)	0.97 (OR 1.0, CI 0.2–5.3)
	During pregnancy	0.25 (OR 1.9, CI 0.7–5.5)	0.73 (OR 0.7, CI 0.1–6.4)	N.T.	0.72 (OR 1.4, CI 0.2–8.8)	0.26 (OR 2.1, CI 0.6–7.5)
**APS before pregnancy**		0.12 (OR 3.1, CI 0.8–12.7)	0.28 (OR 2.6, CI 0.5–13.9)	**0.04 (OR = 3.3, CI 1.1–10.2)**	0.22 (OR 3.5, CI 0.5–26.2)	0.13 (OR 3.6, CI 0.7–18.5)
**Previous LN**		**0.005 (OR = 5.9, CI 1.7–20.8)**	0.06 (OR 5.7, CI 0.9–34.6)	0.20 (OR 2.6, CI 0.6–10.7)	0.10 (OR 7.0, CI 0.7–71.5)	**0.01 (OR = 16.6, CI 1.8–156.5)**

* IUGR, (intra-uterine growth restriction) and/or; SGA (small-for-gestational-age) newborns and/or; LBW, (low birth weight). Yellow background indicates statistical significance. aCL, anti-cardiolipin antibody; anti-β2-GPI, anti-β2-glycoprotein I antibody; APS, antiphospholipid syndrome; CI, 95% confidence intervals; LA, lupus anticoagulant test; LN, lupus nephritis; N.E.; not estimated (used for calculations with division by zero); N.T., not tested; OR, odds ratio.

**Table 6 jcm-09-02598-t006:** Logistic regression analysis with associations between risk factors and development of any APO.

Risk Factors	Univariate			Multivariable		
	OR	CI	*p*-Value	OR	CI	*p*-Value
**APO in all pregnancies** (*n* = 59)						
Previous LN	**5.9**	**1.9–18.8**	**0.002**	**6.0**	**1.3–27.9**	**0.02**
≥1 pos. aCL, anti-β2-GPI or LA test (ever)	**3.3**	**1.0–10.7**	**0.05**	2.8	0.6–13.9	0.20
≥1 pos. aCL, anti-β2-GPI or LA test (≤1-year before)	0.8	0.2–2.8	0.75	0.12	0.01–1.3	0.09
≥1 pos. aCL, anti-β2-GPI or LA test (during pregnancy)	1.9	0.5–6.6	0.32	2.6	0.2–30.4	0.45
**APO in subsequent pregnancies** (*n* = 33)						
Previous LN	**12.0**	**2.0–72.4**	**0.007**	**25.7**	**1.8–362.3**	**0.02**
≥1 pos. aCL, anti-β2-GPI or LA test (ever)	**11.3**	**1.2–109.3**	**0.04**	15.3	0.71–329.9	0.08
≥pos. aCL, anti-β2-GPI or LA test (≤1-year before)	1.3	0.26–5.9	0.78	0.3	0.008–11.6	0.52
≥pos. aCL, anti-β2-GPI or LA test (during pregnancy)	1.3	0.26–5.9	0.78	0.3	0.008–11.6	0.52

aCL, anti-cardiolipin antibody; anti-β2-GPI, anti-β2-glycoprotein I antibody; CI, 95% confidence intervals; LA, lupus anticoagulant test; LN, lupus nephritis; OR, odds ratios.

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
