# Peer review of "Adverse Pregnancy Outcomes after Multi-Professional Follow-Up of Women with Systemic Lupus Erythematosus: An Observational Study from a Single Centre in Sweden"

_jcm, 2020, doi:10.3390/jcm9082598_

Round 1

Reviewer 1 Report

Dear Authors, 

I read this study with interest. I do not have many concerns, this study provides a good data point for a clinically meaningful topic, although similar studies have already been done.

There are many tables in this paper - this makes it partially difficult to read. You should consider moving some of lesser importance into the Supplement, while a condensed core should remain. This will also make it easier for an analysis and future reference of your study by other researchers.

You mention that this was a prospective trial with written consent to prospective follow-up, and the trial duration is between 2002-2018. Meanwhile, the Ethical acceptance is termed "(Dnr 2010/668)", (I assume this means it was obtained in 2010). Please clarify.

Regards.

Author Response

I read this study with interest. I do not have many concerns, this study provides a good data point for a clinically meaningful topic, although similar studies have already been done.

There are many tables in this paper - this makes it partially difficult to read. You should consider moving some of lesser importance into the Supplement, while a condensed core should remain. This will also make it easier for an analysis and future reference of your study by other researchers.

Reply: Thanks for the overall positive response. In the revised manuscript version, two tables (previous Table 6 and 7) were moved to Supplements.

You mention that this was a prospective trial with written consent to prospective follow-up, and the trial duration is between 2002–2018. Meanwhile, the Ethical acceptance is termed "(Dnr 2010/668)", (I assume this means it was obtained in 2010). Please clarify.

Reply: This was observant. In fact, we have two valid ethical approvals for this project but as the first one is older it is more correct to announce the approval from 2002. This has been changed in the revised manuscript version.

Reviewer 2 Report

The authors have made great efforts to analyze their data. Unfortunately there is nothing really new in the findings and the conclusions.

It is a pity that they did not investigate the effects of medication on APO.

Also the small number of cases in a single center is a major drawback since there are many other comparable studies with much higher numbers of cases.

Author Response

The authors have made great efforts to analyze their data. Unfortunately, there is nothing really new in the findings and the conclusions.

It is a pity that they did not investigate the effects of medication on APO.

Also the small number of cases in a single center is a major drawback since there are many other comparable studies with much higher numbers of cases.

Reply: We appreciate the reviewer’s comments regarding our efforts to put all data together. We agree that the data presented are mainly confirmatory. Unfortunately, we did not have access to data on longitudinal medication (or statistical power to analyze the effects of medication on APO development) and this fact was also mentioned as a limitation. We included all pregnancies from SLE patients in the catchment area of Lund University hospital during 2002–2018, and we did not have ethical approval for a multicentre study. We have made changes in the Results section to present data more clearly.

Reviewer 3 Report

Methods:

  1. SLE criteria: Did the patient also fulfill the more updated version (1997) of the ACR criteria? Hochberg MC. Updating the American College of Rheumatology revised criteria for the classification of systemic lupus erythematosus. Arthritis Rheum. 1997;40(9):1725.
  2. Please show median values for demographics table, since the data seems to be not normally distributed.
  3. Please clarify how the Anti-phospholipid antibodies were present in 17 patients but the APS was diagnosed in 19 patients?
  4. Please mention the standard of care regimen followed for pregnant patients with APS, since very low number of patients were on LMW Heparin or ASA despite 32.2% of patients with APS diagnosis?
  5. Data presented in Table 3 is confusing and difficult to follow as some of it is shown as N=44 and other as N=59. Please either use consistent number or split the table in sperate tables.
  6. In the risk factors section please show the actual data along with OR and p-value. Again, the tables are very confusing and difficult to follow. There are different numbers listed on the top of columns without explanation.
  7. Please add an explanation in the main manuscript about the data showed as NE (not estimated)?
  8. For data in table 8 did you do analysis for APO in first pregnancy?
  9. Please revise figure 1. The Y-axis is labeled as SLEDAI 2K or SDI, these are two different measures and you cannot show them on a single axis.
  10. Please revise this statement in discussion to more accurately express the data. The global estimates for pre-eclampsia is 4.6% and for eclampsia, it is 1.4%. While pre-eclampsia occurs in 2–8% of pregnancies in the general population, and is seldom complicated by eclampsia and the HELLP  syndrome, these maternal complications( you are only referring to pre-eclampsia?) occurred in >20% of the pregnancies investigated in the present study, which is comparable to the results reported in previous investigations.
  11. There was no data shown about the risk factors for the first pregnancy. So, this statement is not entirely accurate: Among the risk factors assessed in this study, a history of LN, diagnosis of APS, and the presence of one or more aPL were associated with major APO, regardless of pregnancy sequence.
  12. The number of patients with general risk factors for APO was extremely low in the study, especially when compared to what has been reported by others. This statement in the discussion is not accurate: Effects of other general risk factors for APO (i.e. smoking, ≥35 years of age at time of conception, or obesity) did not fall out significant.

Author Response

Methods:

1) SLE criteria: Did the patient also fulfill the more updated version (1997) of the ACR criteria? Hochberg MC. Updating the American College of Rheumatology revised criteria for the classification of systemic lupus erythematosus. Arthritis Rheum. 1997;40(9):1725.

Reply: Apart from the 2012 SLICC criteria, all included patients fulfilled the 1997 ACR classification criteria. This information has been included in the revised text (page 2, line 113).

2)Please show median values for demographics table, since the data seems to be not normally distributed.

Reply: This has been added in the revised text in Table 1.

3) Please clarify how the Anti-phospholipid antibodies were present in 17 patients but the APS was diagnosed in 19 patients?

Reply: We admit that the information in Table 1 may cause misunderstanding. In fact, 19 of 59 pregnancies originated from women with a previous diagnosis of APS. However, altogether 17 of 28 women included in the study fulfilled the SLICC-12 antiphospholipid antibody criteria with/without a diagnosis of APS prior to pregnancy. We made efforts to clarify this further in Table 1.

4) Please mention the standard of care regimen followed for pregnant patients with APS, since very low number of patients were on LMW Heparin or ASA despite 32.2% of patients with APS diagnosis?

Reply: Our investigation did not focus in detail on the prescribed pharmacological treatments. In general terms, the treatment of patients with APS was chosen on an individual level. The treatment with ASA and/or LMWH was based on the patients’ clinical phenotype, assessed at the regular visit before conception or at the time of pregnancy counselling. The lack of universal consensus and international guidelines for standard of care regimen in the past years may have contributed to a lower rate of patients on treatment with ASA or LMWH than expected when the current guidelines are followed, especially in the beginning of the millennium.

5) Data presented in Table 3 is confusing and difficult to follow as some of it is shown as N=44 and other as N=59. Please either use consistent number or split the table in sperate tables.

Reply: The reason to why different denominators were used is dependent on early or late APO. For instance, we cannot count “stillbirths” for the pregnancies which ended early due to miscarriage. As stated in the heading of Table 3, we employed pregnancy number as denominator in the case of early outcome or the number of deliveries in the case of late outcome.

6) In the risk factors section please show the actual data along with OR and p-value. Again, the tables are very confusing and difficult to follow. There are different numbers listed on the top of columns without explanation.

Reply: We agree that data in the Tables could be presented more clearly. We now indicate the number of pregnancies that developed each complication under each type of APO. The 2nd row in Table 5 and Supplementary Tables 1+2 (previous Table 6 and 7) shows the dominators, i.e. the total number of examined pregnancies. For instance, regarding early APO (such as miscarriage) we employed all pregnancies whereas late APO (such as preterm delivery) we divided by the number of deliveries.

7) Please add an explanation in the main manuscript about the data showed as NE (not estimated)?

Reply: N.E. (not estimated) was used for calculations with division by zero. N.T. (not tested) was used for instance in comparisons with “miscarriage ≤10 weeks” and risk factors “during pregnancy”. N.E. was clarified with foot notes under Table 5 (and previous Table 6 and 7 which were moved to Supplements in the revised version). Further clarification is given under subheading “Risk factors” at the Results section (page 7, line 372-373).

8) For data in table 8 did you do analysis for APO in first pregnancy?

Reply: None of the risk factors fell out significant in the univariate analysis for 1st pregnancies versus total APO. Thus, we decided to only show data for “all” and “subsequent” pregnancy groups. This has been clarified (page 10 line 450-452).

9) Please revise figure 1. The Y-axis is labeled as SLEDAI 2K or SDI, these are two different measures and you cannot show them on a single axis.

Reply: Figure 1 has been revised and now also includes a right Y-axis.

10) Please revise this statement in discussion to more accurately express the data. The global estimates for pre-eclampsia is 4.6% and for eclampsia, it is 1.4%. While pre-eclampsia occurs in 2–8% of pregnancies in the general population, and is seldom complicated by eclampsia and the HELLP  syndrome, these maternal complications( you are only referring to pre-eclampsia?) occurred in >20% of the pregnancies investigated in the present study, which is comparable to the results reported in previous investigations.

Reply: This statement has been revised (page 11, line 539).

11) There was no data shown about the risk factors for the first pregnancy. So, this statement is not entirely accurate: Among the risk factors assessed in this study, a history of LN, diagnosis of APS, and the presence of one or more aPL were associated with major APO, regardless of pregnancy sequence.

Reply: Thanks for this comment. It is correct that the multivariable analysis was not done separately for the 1st pregnancy group (see above). Thus, we have changed the sentence and excluded the last part.

12) The number of patients with general risk factors for APO was extremely low in the study, especially when compared to what has been reported by others. This statement in the discussion is not accurate: Effects of other general risk factors for APO (i.e. smoking, ≥35 years of age at time of conception, or obesity) did not fall out significant.

Reply: Thanks for this comment. It is correct that the percentage of patients with general risk factors was low, but it reflects the situation in Sweden. We agree, however, that this also limits the possibility to draw firm conclusions regarding general risk factors. Thus, this statement has been revised (page 12, line 606-607).

Reviewer 4 Report

In the present manuscript, the authors evaluated the occurrence of adverse pregnancy outcomes (APO) in a longitudinal study during in 2002–2018 among patients with SLE from the catchment area of the Department of Rheumatology in Lund, Sweden. Despite the interest topic analyzed in this manuscript, the novelty of the data seems to be weak and the number of patients included in the study is very low compared to previous published articles in this area.

The two principal weakness of the data presented are the lack of analysis of the effect of medication on the APO and the small number of studied pregnancies, as authors indicated as limitations of the study. The small size cohort influence on low statistical significance and the large OR confidence intervals of the risk factors analyzed, suggesting a little statistical power of the results.

Other weakness of the study is the little novelty of the analysis, perhaps additional potential biomarkers should be evaluated to identify high and new risk factors of APO. For example, hypertension o hypertension disorders during pregnancy is an established risk factor of APO (fetal loss, prematurity and intrauterine growth restriction) in SLE and also the lupus flares, which should be included in the analysis as a potential risk factor of APO in this study. On the other hand, a recent study (Kim et al, Am J Obstet Gynecol 2016) pointed angiogenic factors (sFlt1, PlGF, and soluble endoglin) as predictors of APO in high-risk SLE and/or APL pregnancies, which can also be analyzed.

Another point, disease activity by SLEDAI-2K in the cohort was measured and showed in Figure 1, indicating a modest increase in disease activity during the last two trimesters and the near period post-delivery. But, do lupus flares also noted during follow-up? Is there any lupus flare?

The term "subnormal levels of C3" employed by authors in page 7, lane 179, is not adequate, it should be replace by low levels of C3.

The conclusions are audacious for the results presented. The slight significance obtained in risk factor analysis is not enough to concluded that previous nephritis is the main feature associated with APO in SLE. In addition, the association of previous LN as a risk factor of APO is based on an “ever documented LN”, and I think is insufficient. The LN criteria should be more explained in detail in the Risk Factors description. Then, in results section authors said “history of renal involvement” (page 7, line 174), What want the authors to say? Is it different from previous LN?

Finally, the results did not provide important clinical information to be applied.

Author Response

In the present manuscript, the authors evaluated the occurrence of adverse pregnancy outcomes (APO) in a longitudinal study during in 2002–2018 among patients with SLE from the catchment area of the Department of Rheumatology in Lund, Sweden. Despite the interest topic analyzed in this manuscript, the novelty of the data seems to be weak and the number of patients included in the study is very low compared to previous published articles in this area.

Reply: We appreciate the reviewer’s comments regarding interest of the topic. We agree that the data presented are mainly confirmatory although similar data from Scandinavia have not previously been published to our knowledge. We included all pregnancies from SLE patients in the catchment area of Lund University hospital during 2002–2018, but we did not have ethical approval for a multicentre study.

The two principal weakness of the data presented are the lack of analysis of the effect of medication on the APO and the small number of studied pregnancies, as authors indicated as limitations of the study. The small size cohort influence on low statistical significance and the large OR confidence intervals of the risk factors analyzed, suggesting a little statistical power of the results.

Reply: Unfortunately, we did not have access to data on longitudinal medication and were thus not able to analyze its effect on development of APO. This constitutes a limitation which we mention in the end of the Discussion along with other limitations, such as small study population yielding low statistical power.

Other weakness of the study is the little novelty of the analysis, perhaps additional potential biomarkers should be evaluated to identify high and new risk factors of APO. For example, hypertension o hypertension disorders during pregnancy is an established risk factor of APO (fetal loss, prematurity and intrauterine growth restriction) in SLE and also the lupus flares, which should be included in the analysis as a potential risk factor of APO in this study. On the other hand, a recent study (Kim et al, Am J Obstet Gynecol 2016) pointed angiogenic factors (sFlt1, PlGF, and soluble endoglin) as predictors of APO in high-risk SLE and/or APL pregnancies, which can also be analyzed.

Reply: We appreciate the reviewer’s comments regarding “new” biomarker analyses in relation to outcomes of SLE pregnancies. This is indeed an appealing approach which we nevertheless consider to be outside the scope of the present manuscript. Please note that established SLE biomarkers, i.e. ANA subspecificities (Ro/SSA, La/SSB, anti-dsDNA), antiphospholipid antibodies and complement proteins were analyzed in all pregnancies at different time-points.

Another point, disease activity by SLEDAI-2K in the cohort was measured and showed in Figure 1, indicating a modest increase in disease activity during the last two trimesters and the near period post-delivery. But, do lupus flares also noted during follow-up? Is there any lupus flare?

Reply: Thank you for this important comment. In this study, we defined “active SLE” as a SLEDAI-2K score of ≥4 (page 2, line 132-133). During the 6-months pre-conception period, active SLE was observed in 12/42 (29%). During the 1st trimester, active SLE was observed in 12/40 (30%). During the 2nd trimester, active SLE was observed in 19/39 (49%). During the 3rd trimester, active SLE was observed in 17/40 (43%). During the 6-months post-partum period, active SLE was observed in 22/43 (51%). This important information was added to the revised manuscript (page 10, line 461-465).

The term "subnormal levels of C3" employed by authors in page 7, lane 179, is not adequate, it should be replace by low levels of C3.

Reply: This has been corrected (page 7, line 361).

The conclusions are audacious for the results presented. The slight significance obtained in risk factor analysis is not enough to concluded that previous nephritis is the main feature associated with APO in SLE. In addition, the association of previous LN as a risk factor of APO is based on an “ever documented LN”, and I think is insufficient. The LN criteria should be more explained in detail in the Risk Factors description. Then, in results section authors said “history of renal involvement” (page 7, line 174), What want the authors to say? Is it different from previous LN?

Reply: “Previous LN” in this study was defined as ever having met the ACR criterion for “renal disorder” (n=15). In addition to fulfilment of the ACR criterion, the diagnosis was based on renal histopathology compatible with LN in 15 of 15 patients. This information was added (page 3, line 206). The statement of “history of renal involvement” is misleading and has been changed (page 7, line 367; page 11, line 535) to LN, or history of LN.

Finally, the results did not provide important clinical information to be applied.

Reply: We admit to the limitations of the study, but we have provided all clinical information available on the material and believe our experiences of long-term follow-up of a specific population may be of interest.

Round 2

Reviewer 4 Report

First of all, thanks to authors for considering and including some of the my suggestions in the revised manuscript.

However, the most important critical aspects of my revision are not adressed in the revised manuscript, becuase the authors do not have the clinical information or permissions.

Therefore, although the authors have added new data, the manuscript is still not very novel, and conclussions continue being ambicious, depending of the results presented.

Author Response

Reviewer-4

First of all, thanks to authors for considering and including some of my suggestions in the revised manuscript.

Reply: Thank you. The suggestions were valuable and have improved the manuscript.

However, the most important critical aspects of my revision are not addressed in the revised manuscript, because the authors do not have the clinical information or permissions.

Reply: Yes, correct. We have already provided all available clinical and laboratory information and data. We emphasize limitations, including the small study population, in the paper. To include further pregnancies at this point is not possible.

Therefore, although the authors have added new data, the manuscript is still not very novel, and conclusions continue being ambicious, depending of the results presented.

Reply: We agree that firm conclusions of risks for APO among those with prior lupus nephritis cannot be made based on this limited study alone. In the second revision, we are more cautious regarding our conclusions. We have edited the manuscript title, as well as some sentences in the abstract, discussion and conclusions. We acknowledge that the data presented are mainly confirmatory and we have clearly stated in ‘Conclusions’ that "no new predictors could be identified in this study". Nevertheless, the novelty of our study concerns the long-term real-world data, gathered over a period of nearly two decades in a single tertiary centre, where a multidisciplinary team has followed patients with SLE over time with similar strict guidelines in a contest of low selection bias due to the universal access to the public, tax-funded nature of the Swedish healthcare system. As this kind of study have not previously been performed in Scandinavia, we believe that our contribution would be relevant to validate, in a Scandinavian population, the role of predictors of APO already highlighted in larger studies from other countries. We hope this proves sufficient for a favourable editorial decision.